# Natural Bred ε^2^-Phages Have an Improved Host Range and Virulence against Uropathogenic *Escherichia coli* over Their Ancestor Phages

**DOI:** 10.3390/antibiotics10111337

**Published:** 2021-11-01

**Authors:** Maria Loose, David Sáez Moreno, Michele Mutti, Eva Hitzenhammer, Zehra Visram, David Dippel, Susanne Schertler, Lenka Podpera Tišáková, Johannes Wittmann, Lorenzo Corsini, Florian Wagenlehner

**Affiliations:** 1Clinic for Urology, Pediatric Urology and Andrology, Justus-Liebig University Giessen, 35392 Giessen, Germany; ml.85@gmx.net (M.L.); David.Dippel@chemie.bio.uni-giessen.de (D.D.); 2PhagoMed Biopharma GmbH, A-1110 Vienna, Austria; david_gamonal95@hotmail.com (D.S.M.); Michele.mutti@phagomed.com (M.M.); eva.hitzenhammer@gmail.com (E.H.); Zehra.visram@phagomed.com (Z.V.); Lenka.tisakova@phagomed.com (L.P.T.); 3DSMZ—German Collection of Microorganism and Cell Cultures GmbH, Leibniz Institute, 38124 Braunschweig, Germany; sus18@dsmz.de (S.S.); jow12@dsmz.de (J.W.)

**Keywords:** phage therapy, phage breeding, *E. coli*, urinary tract infections, phage training, homologous recombination

## Abstract

Alternative treatments for *Escherichia coli* infections are urgently needed, and phage therapy is a promising option where antibiotics fail, especially for urinary tract infections (UTI). We used wastewater-isolated phages to test their lytic activity against a panel of 47 *E. coli* strains reflecting the diversity of strains found in UTI, including sequence type 131, 73 and 69. The plaquing host range (PHR) was between 13 and 63%. In contrast, the kinetic host range (KHR), describing the percentage of strains for which growth in suspension was suppressed for 24 h, was between 0% and 19%, substantially lower than the PHR. To improve the phage host range and their efficacy, we bred the phages by mixing and propagating cocktails on a subset of *E. coli* strains. The bred phages, which we termed evolution-squared ε^2^-phages, of a mixture of *Myoviridae* have KHRs up to 23% broader compared to their ancestors. Furthermore, using constant phage concentrations, *Myoviridae* ε^2^-phages suppressed the growth of higher bacterial inocula than their ancestors did. Thus, the ε^2^-phages were more virulent compared to their ancestors. Analysis of the genetic sequences of the ε^2^-phages with the broadest host range reveals that they are mosaic intercrossings of 2–3 ancestor phages. The recombination sites are distributed over the whole length of the genome. All ε^2^-phages are devoid of genes conferring lysogeny, antibiotic resistance, or virulence. Overall, this study shows that ε^2^-phages are remarkably more suitable than the wild-type phages for phage therapy.

## 1. Introduction

Uncomplicated urinary tract infections (UTI), which are caused predominantly by uropathogenic *Escherichia coli* (UPEC) are amongst the most common reasons for medical consultation [1]. UPEC strains show different constellations of virulence factors and diverse genetic backgrounds. Several international studies performing DNA profiling revealed the predominance of the sequence types (ST) 69, 73, 95, and 131, among others, in isolates from patients with UTI [2,3,4,5,6,7]. These STs differ markedly in their antibiotic resistance profiles. While STs 69, 73, and 95 are largely susceptible except rarely extended-spectrum β-lactamases (ESBLs), ST131 isolates often carry ESBLs together with fluoroquinolone and aminoglycoside resistance [8]. Oral antibiotics are preferred for the treatment of mild and moderate uncomplicated UTI [9]. However, due to the increasing rates of resistance of uropathogens against cotrimoxazole, fluoroquinolones, and β-lactams, these classical oral antibiotics cannot be used for empiric therapy any more in many geographic regions [10,11]. To counteract the increasing resistance rates, alternative treatment options need to be investigated.

One promising avenue is the use of bacteriophages (hereinafter referred to as phages). The first reported application of therapeutic phages can be traced back to the early 20th century [12], where phages were used to control a variety of diseases including diarrhea, cholera, dysentery, and salmonellosis [13]. Nevertheless, this idea was pursued in the Soviet Union, but largely ignored in the Western world, as antibiotics were considered superior [14]. The advantage of phages over antibiotics is their specificity towards bacterial species or even strains, preventing the collateral effects on commensal gut microbes and the associated side effects. However, this same specificity limits the ability of a particular phage’s use to a small set of potential pathogens requiring more specific diagnosis. Furthermore, phage virulence must be high enough to decrease bacterial population efficiently in vivo [15,16] and simultaneously prevent the selection of phage-insensitive subpopulations [17,18]. The inability to cover all strains of a certain species, along with the development of evolutionary resistance by bacteria against their phages, have restricted phage therapy or phage biocontrol thus far. Wide host range and high virulence are, therefore, two desired traits for therapeutic phages.

To increase the therapeutic efficacy of phages, multiple breeding methods have been proposed. Direct manipulation of the viral genome, “genetic breeding” [19], allowed the increase of the phage absorption to their host by mutating tail spike proteins or the conversion of temperate phages to obligate lytic ones [20,21]. However, genetic breeding is limited by the understanding of the phage genomes and, so far, most of their encoded genes are often of unknown function [22]. Therefore, different breeding strategies based on random changes of the genomes have been developed. One of these strategies focuses on exploiting natural selection and directed evolution of phages, known as the Appleman’s Protocol [23]. It is based on exposing phage cocktails to multiple rounds of infection of bacteria and selecting the evolved phage progeny with advantageous traits [23,24]. Multiple phages infecting a bacterial cell can undergo mutations and random intercrossing of whole genomic regions by homologous recombination that result in improved phages. The principle of ‘Cross-breeding’ was already described in 1947 for *E. coli* phages with different plaque morphologies [25], and in nature, genetic intercrossing is one of the drivers of phage evolution, which leads to the mosaicism of phage genomes [26]. Some authors have used this method to generate phages with a wider host range [27,28] and we recently described the optimization of this method to generate *Staphylococcus aureus* phages with a vastly improved virulency and host range [29].

*E. coli* infecting phages are readily isolated from a variety of sources with approximately 600 genomes sequenced to date. They are represented in several phage families including *Leviviridae*, *Siphoviridae*, *Podoviridae*, *Myoviridae*, and *Microviridae*. Currently, within these families, there are 37 genera and 157 species that contain *E. coli* infecting phages, with the many taxa being poorly sampled [30]. More than 20 years ago, the efficiency of multispecies phage cocktails as a treatment for acute and chronic urogenital inflammation was already described, but the study is poorly documented for use as a model for human phage therapy [31]. The Myovirus KEP10, showing bacteriolytic activity against UPEC strains and stability in human and murine urine, diminished death mediated by UPEC-induced UTI in mice [32] and the treatment of UTI patients with phage cocktails has been reported [33]. The lytic activity of this commercial phage cocktail was increased by ‘bacteriophage adaptation’ experiments. Although successful, no characterization of the improved phages was provided [34], which could pose a challenge towards future regulatory approval [35]. A recent study on phage lambda demonstrated how phage-host co-evolution led to a phage receptor expansion and lower resistance rate [36]. So far, other phage adaptation studies, including the co-culturing of phages T4 and T7 with resistant strains, have generated phage mutants that can switch to utilize alternative receptors when under selection pressure [37,38,39]. Nevertheless, the determination for an increased host range of these mutants is sparse. Gibson et al. (2019) also claimed a successful host range expansion for *E. coli* phages, although the study does not provide supporting data [27]. Kuniska & Tanji (2010) showed the recombination of phage genomes after infecting one *E. coli* strain with a cocktail of three phages but did not compare the efficacy of these with their ancestors [40].

In this study, we isolated new *E. coli* phages from wastewater and showed their improvement towards therapeutic use in UTI treatment by intercrossing different wild-type phages by superinfection, and by selecting the best progeny (ε^2^-phages), which showed an improved host range and increased virulence.

## 2. Results

### 2.1. Isolation of Phages from Waste Water Reveales 28 Novel Phages

Several lytic phages were isolated from wastewater samples. Twenty-eight phages showing clear plaques with activity against *E. coli* strains were selected for further characterization. Sequence analysis revealed that all these phages belonged to the order Caudovirales, tailed phages having double stranded DNA genomes [41]. The most predominant family among the sequenced phages was *Autographiviridae* (*n* = 11), followed by *Myoviridae* (*n* = 7), *Siphoviridae* (*n* = 7), and *Podoviridae* (*n* = 3). At the subfamily and genus level, *Autographiviridae* phages belonged to the subfamily *Molineuxvirinae* (*Vectrevirus* or unspecified genus), and *Studiervirinae* (*Kayfunavirus* genus). Phages from the *Myoviridae* family belonged to the *Tevenvirinae* subfamily, the majority characterized as members of the genus *Tequatrovirus* and one belonging to the genus *Mosigvirus*. *Siphoviridae* phages were classified into subfamily *Guernseyvirinae* (*Kagunavirus* genus) and genus *Dhillonvirus*, while all podoviruses belonged to the *Kuravirus* genus (Table 1).

To validate the classification achieved by sequence comparison, selected phages were additionally analyzed by transmission electron microscopy (TEM). The phages CHD94UKE2, 101117BS1, and CHD16UKE1, classified as *Tequatrovirus*, showed a long contractile tail with short spikes characteristic for the family *Myoviridae* (Figure 1).

The morphology of phage 101120B2 showed an isometric phage with a short, non-contractile tail as expected for the family *Autographiviridae*. Phage 101114BS3 was assigned to *Kuravirus* of the *Podoviridae* by sequence comparison, and its TEM picture revealed an elongated capsid and a short tail, previously seen in other *Podoviridae* [30]. The TEM picture of phage 22664BS2, belonging to the genus *Kagunavirus* of the *Siphoviridae* family, showed the characteristic long, non-contractile tail (Figure 1).

### 2.2. The Plaquing Host Range Does Not Correlate with the Ability of Phages to Control Growth in Suspension

To gain a first insight into the efficiency of the isolated phages, the susceptibility of a panel of 79 *E. coli* strains, isolated from urinary tract infections (Appendix A), was tested. Here, the formation of clear plaques was defined as phage susceptibility of the host strain. Overall, the plaquing host range (PHR) ranged from 61% (*Autographiviridae* 22664UKE3-2) to 1% (*Siphoviridae* UTI89UKE1) (Table 1).

For further characterization of the host range of phages intended for the treatment of UTIs, we compiled a panel of 47 *E. coli* strains reflecting the most prominent sequence types of uropathogenic *E. coli* (UPEC), including antibiotic resistant and multiresistant strains (Table 2 and Appendix A).

We selected phages with a promising host range for further characterization on the UPEC panel including the myoviruses CHD16UKE1, CHD94UKE2, G3G7, and 101117BS1, as well as members of the *Autographiviridae*, 22664UKE3-2, 101120B2, and 101118B1. The efficiency of plaquing (EOP) of each of the seven phages was determined against the UPEC panel. The plaquing host range (PHR) was defined as the % of strains where clear plaques were formed. Strains on which the phage spot locally impaired growth (“opaque lysis”), but no clear plaque was formed, were not counted for the PHR. The myovirus G3G7, as well as the *Autographiviridae* 22664UKE3-2, showed the highest PHR with each 49% followed by CHD16UKE1 (43%), CHD94UKE2 (34%), 101118B1 (16%), 101120B2 (13%), and 101117BS1 (9%) (Figure 2).

In addition, we tested to which extent the bacterial growth in suspension could be impaired by the phages. Therefore, we measured what we termed, kinetic host range (KHR). The KHR is defined as the number of strains showing less than 10% growth in suspension after phage treatment compared with their untreated control. The percentage of growth was determined by OD600. The KHR values were remarkably lower than the PHRs for every phage, ranging between 0% (101118B1) and 20% (CHD16UKE1) (Figure 1). Phages G3G7 and 22664UKE3-2 showed a particularly large difference between the PHR and KHR, with a reduction of the host range by approximately 94 percentage points and 76 percentage points, respectively, in the KHR compared to PHR (Figure 2).

To analyze the correlation between the EOP and the ability to inhibit the growth of bacteria in suspension after 24 h, we defined the OD ratio as the ratio of optical densities of the phage-treated and untreated bacterial suspensions. Anticorrelation is expected in this analysis, i.e., that growth in suspension (quantified as OD ratio) is inhibited best on strains where the EOP is high. However, this was the case only for a few bacterial strains (Figure 2). As depicted in Figure 2b,c, the correlation was poor. All phages failed to inhibit bacterial growth in suspension for 24 h even on strains where they have a high EOP.

### 2.3. Breeding of Myoviridae Phages Improves Their Kinetic Host Range against Uropathogenic E. coli 

To improve the host range of the wild-type phages, mixtures of two different sets of phages were bred by incubation with 24 *E. coli* strains (Appendix A) in artificial urine for 20 rounds, essentially as described [29]. The first breeding cocktail contained the myoviruses CHD16UKE1, CHD94UKE2, 101117BS1, and G3G7. A second breeding cocktail included the three *Autographiviridae* 22664UKE3-2, 101120B2, and 101118B1. The host range of the single phages isolated after breeding (ε^2^, or evolution squared phages) was determined by plaquing and by the OD ratio on the panel of 47 UPEC strains described above (Table 2). The PHR of the ε^2^ myoviruses ranged from 47–51%, which was not improved over the best ancestor G3G7, but higher than the ancestors with the best KHR, CHD16UKE1 (PHR = 43%) and CHD94UKE2 (PHR = 34%). Strikingly, four of the six ε^2^ phages with morphology of the *Myoviridae* tested showed an improvement of 18–23% in KHR compared to their most efficient ancestor phage, CHD16UKE1 (Figure 3).

Therefore, the ε^2^
*Myoviridae* seem to have acquired the best combination of properties with regards to maximizing both the PHR and KHR. This effect is further visualized in Figure 3b, showing that the bred phages had both a higher KHR and PHR compared to the ancestors. In addition, for these four ε^2^
*Myoviridae* phages, we analyzed the correlation between the EOP and OD ratio after 24 h. As depicted in Figure 3c, the correlation of the EOP and OD ratio was still limited, but low OD ratios were reached for more strains compared to the ancestors. This directly results in a higher KHR, as shown in Figure 3a,b.

In the breeding group of *Autographiviridae*, ε^2^-phages showed no improvement at either plaquing or kinetic host range over their ancestors (Appendix A).

Next, we analyzed whether the activity of the phages could be linked to the phylogenetic relationships of the UPEC strains. All *Myoviridae* ancestor phages were isolated from wastewater by propagation on various *E. coli* strains of the sequence type 131, one of the most prevalent UPECs. The host range data of the individual tested *E. coli* strains revealed that the phages were most active on strains and sequence types within the B2 phylogenetic group to which the ST131 belongs (Figure 4).

Furthermore, multiple *E. coli* of the phylogenetic groups A and D, less so B1, were susceptible to the *Myoviridae* phages. Here, not all representative strains tested of a phylogenetic group or sequence type showed the same susceptibility (Figure 4). Comparison of the kinetic host range data after 6 h and 24 h incubation revealed that the phages could initially inhibit the growth of up to 60% of the bacteria tested, comparable to the plaquing host range. Nevertheless, after 24 h this number dropped distinctly suggesting a putative resistance evolution of the bacteria (Figure 4).

### 2.4. Myoviridae ε^2^-Phages Are More Virulent than Their Ancestors

We further analyzed whether the breeding process, in addition to expanding the host range, also improved the virulence, or killing efficacy, of the ε^2^-phages over their ancestors. We examined the growth of a subset of 22 of the 47 UPEC strains (Appendix A) in the presence of selected ε^2^-phages at different multiplicities of infection (MOI). This subset panel included 12 strains of the most prevalent phylogenetic group B2 and is further described in Appendix A. The phage titer was kept constant at 1 × 10^7^ PFU/mL and was used to infect bacterial suspensions with MOI 100–0.1. At MOI 100, the highest MOI tested, the *Myoviridae* ancestors could only inhibit the growth of 30–50% of the strains used. In contrast, the ε^2^-phages restrained the growth of 55–79% of the strains at MOI 100 (Figure 5).

Reducing the MOI decreased the overall number of bacteria whose growth could be restricted. Nevertheless, the ε^2^-phages, especially PM123, were more efficient compared to their ancestors with the same MOI. Furthermore, the efficacy of PM123 at MOI 1 (31% of strains) was comparable to the efficacy of the ancestors at MOI 10 (range 17–33% of strains). At MOI 10, PM123 was even more efficient in inhibiting bacterial growth (62% of strains) than any of the ancestors at MOI 100 (range 30–50% of strains) (Figure 5). These results show an increase of phage virulence after the breeding, in the panel of bacterium tested.

### 2.5. The Genomes of ε^2^-Phages Are Intercrossed from Up to 3 Ancestors

The ε^2^-phages resulting from the *Myoviridae* breeding were classified as *Tequatrovirus* by sequence similarity. Each of the *Myoviridae* ε^2^-phages showed extensive genomic recombinations, and genome stretches were inherited from up to four different ancestor phages. The backbone of all *Myoviridae* ε^2^-phages consists largely of the ancestors CHD16UKE1 (A in Figure 6a) and G3G7 (B in Figure 6), which were also the ancestor phages with the highest KHR.

As the identity of CHD16UKE1 and G3G7 was 99.5%, several genomic stretches of the progeny were identical to both (represented as “AB” in Figure 6). Stretches of ancestors CHD94UKE2 and 101117BS1 were distributed differently along the genomes of the progeny. In PM123 and PM136, genomic stretches from the ancestor CHD94UKE2 comprised 11% and 7% of their total coding regions, respectively, and the ancestor 101117BS1 comprised only 1% in both cases. In contrast, the ancestor CHD94UKE2 made up 1 to 2% and 101117BS1 2–8% of the genomes of PM105, PM107, PM108, and PM133 (Figure 6).

A more detailed analysis of an ~10 kb genomic region (protein coding sequence (CDS) 117 to 133) of phage PM123, one of the phages with improved virulence, revealed at least eight recombination sites (Figure 6), where the PM123 genome switches from one ancestor to another. CDS 117 was identical to the ancestors CHD16UKE1 and G3G7, whereas CDS 118 to 123 were identical to the ancestor CHD94UKE2 and CDS 124 was inherited from the ancestor G3G7 (Figure 6). CDS 130 encodes for a tail fiber protein and was inherited from CHD94UKE2. However, it also contained a mutation (highlighted by a green box in Figure 6b), not presented in any of the ancestors, which translated into an amino acid change (Q166R). This confirms that the breeding process not only promotes recombination, but also allows for the occurrence of spontaneous mutations. Moreover, CDS 131, another tail fiber-encoding gene, was identical to ancestor CHD94UKE2 for the first 2.3 kbp, then switched to ancestors CHD16UKE1 or G3G7 (identical in this stretch) for approximately 400 bp, before switching back to CHD94UKE2 for the last 300 bp (Figure 6).

All ε^2^-phages recovered from the *Autographiviridae* mix were classified as *Kayfunavirus* indicating the absence of the positive selection or the dilution of the *Vectrevirus* 101118B1 ancestor during the *Autographiviridae* breeding. The genomic analysis of the ε^2^-phages revealed a distinct mosaicism, where individual genome stretches are inherited from different ancestor phages. The individual *Autographiviridae* ε^2^-phages had a very similar genome arrangement with a backbone inherited from 101120B2. Only CDS 010 could be unambiguously identified as inherited from ancestor 22664UKE3-2, while several CDS might be inherited from either of both phages (101112B2 or 22664UKE3-2, Appendix A). The uniqueness of each *Autographiviridae* ε^2^-phage was defined by different single point mutations, the majority of them accumulated in CDS 001 (putative transglycosylase) and CDS 002 (K5 lyase).

We further analyzed the phage life cycle with PHACTS, screened the genome for antibiotic resistance genes with CARD and ResFinder 4.1, and for virulence factors in Virulence Finder 2.0. All ε^2^-phages (*Myoviridae* and *Autographiviridae*) were devoid of genes conferring lysogeny, antibiotic resistance, or virulence, and were therefore suitable for phage therapy in principle.

## 3. Discussion

Increasing antibiotic resistance rates in UPEC requires alternative treatment options. In this study, we isolated new phages infecting UPEC and improved their therapeutic potential by cross-breeding. We describe for the first time the phage breeding by the principles of the Appleman’s protocol [23] for improving *E. coli* phages towards therapeutic use. 

In order to direct the phage activity towards UPEC, we isolated the phages on strains of sequence type 131, one of the most prevalent in UTI. After an initial characterization, the most promising ones according to host range were selected for breeding. Superinfections of phage cocktails of 3–4 ancestors on a defined set of UPEC strains were used to induce genomic recombination and create bred phages, which we termed ε^2^-phages (evolution squared). The breeding process increased the host range of the ε^2^-phages by 23% compared to the best ancestor and increased their virulence.

Comparing the plaque formation and the killing of planktonic bacteria in suspension revealed significant differences in the determined host range with less efficacy of phages in the kinetic measurements. Recently, we observed this phenomenon for *S. aureus* phages as well [29]. The KHR after 6 h was comparable to the PHR, indicating that initial phage efficiency is comparable in both assays. However, the 24 h KHR determination showed resistance formation of the bacteria against the phages. This rapid regrowth was previously described for *E. coli* and for other Gram-negative bacteria such as *Salmonella* and *Pseudomonas* [43,44,45]. Since the ability of plaque formation does not reflect the phage capability of controlling or reducing bacterial growth, which might be related to the formation of resistance, the use of the “phagogram” [33,46] as the sole predictor of the therapeutic success could be questioned. In fact, some specialized centers are already testing activity in broth, in addition to the traditional spot testing before phage therapy [47].

The *Myoviridae* ancestor and ε^2^-phages were more efficient against ST131 strains or other sequence types belonging to the same phylogenetic group, B2, such as ST73. Other phylogenetic groups such as A and D showed less susceptibility towards the phages. Nevertheless, the pathogenic *E. coli* strains causing extraintestinal infections mainly belong to group B2 and to a lesser extent to group D, whereas commensal strains belong to group A and B1 [48]. Hence, the breeding is successfully driving phage capabilities towards the treatment of UTI infections. The fact that the host range of the ε^2^-phages was increased and not limited to ST131could confer an advantage upon treatment over the narrower host-range of phages previously described [43]. 

Although the panel used for breeding was diverse, the *E. coli* B2-phylogenetic group was predominant, and the selective pressure of this group might have influenced the enhanced virulency towards these strains of *Tequatrovirus* ε^2^-phages, which showed an increase in host range and virulence. Previous studies have pointed to gp37 as the driver of host range expansion in a T4-related phage [49]; however, the number of mutations and recombination found in our ε^2^-phages makes it virtually impossible to identify which genes caused this improvement. Contrarily, Holtzman et al. (2020) showed that phage T7 specializes and contracts its host range towards certain types of LPS if the other LPS-types do not allow productive infection [37]. Our breeding could have had a similar impact in *Autographiviridae* phages, which did not improve their host range. The diverging outcome of the breeding protocol for the different phage families could be due to i) the fact that genetically more similar *Myoviridae*-phages were used in the initial ancestor cocktail, allowing more chances of recombinations or ii) due to the *Myoviridae* larger genome, which might allow more recombinations, compared to the *Autographiviridae* phages. 

Recently, using a similar method, we created natural ε^2^-phages of *S. aureus* showing a host range twice as broad as their ancestors [29]. The fact that the host range improvement in the ε^2^-phages of *E. coli* is not as large could be related to the fact that *E. coli* strains are more diverse. The core genome of *E. coli* comprises only up to 25% of the genes, while the remaining are considered as accessory genomes [50,51,52]. In contrast, approximately 75% of *S. aureus* genes were classified into the core genome [53].

Although phage cocktails were not formulated in this study, the genomic differences of the ε^2^-phages could allow the creation of complementary cocktails. Since adsorption inhibition seems to be the main phage-resistance mechanism identified in *E. coli* [54], the identification of specific phage receptors and subsequently creating cocktails targeting different receptors has the potential to reduce resistance formation and therefore could drive therapeutic success [55]. Furthermore, the breeding was carried out in artificial urine, which might confer an advantage to the ε^2^-phages during UTI treatment.

In conclusion, this study describes ε^2^-phages lysing *E. coli*, bred from newly isolated wild-type phages. Homologous recombination and selection led to phages with superior killing in suspension (KHR) and higher virulence than the wild-type phages. The specialization of these phages towards certain ST types of *E. coli*, independent of its antibiotic resistance, could confer an advantage for the therapeutic treatment of UTI. We show that it is possible to substantially improve key phage properties required for therapeutic success by applying directed evolution with limited knowledge of the underlying functions of the altered parts of the genomes. The ε^2^-phages described in this study may not only have increased therapeutic efficacy compared to the ancestors, but, due to the increased kinetic host range, may have also reduced the number of phages required in a fixed cocktail or in a phage bank for personalized phage therapy.

## 4. Materials and Methods

### 4.1. Bacterial Strains

All bacterial strains used in this study are listed in Appendix A. *E. coli* ATCC25922 was obtained from ATCC (Wesel, Germany) as reference strains for antimicrobial susceptibility testing. *E. coli* UTI89 and 83972 were obtained from the DSMZ (Braunschweig, Germany). The strains *E. coli* CDF1, CDF2, CDF6, CDF8, Af23, Af24, Af31, Af40, Af45, Af48, and S115 were kindly provided by Patrice Nordmann. All other strains were clinical isolates from patients with urinary tract infections. Bacteria were cultivated in Luria-Bertani (LB) broth (Carl Roth, Graz, Austria), tryptic soy broth (TSB) (Thermo Fisher Scientific, Waltham, MA, USA) and artificial urine [56] or on LB and CAMHB (Becton, Dickinson and Company, Franklin Lakes, NJ, USA) agar plates.

### 4.2. Phage Collection and Propagation

Phages were isolated from wastewater samples collected in 2018 from the Charité (Berlin, Germany), the DSMZ (Braunschweig, Germany) and the University Medical Center Hamburg-Eppendorf (Hamburg, Germany) at the Leibniz Institute DSMZ-German Collection of Microorganisms and Cell Cultures.

Briefly, wastewater samples were filtered through a 0.22 µm-pore filter and mixed with several *E. coli* strains grown to log phase (OD 600 nm = 0.4). The mixture was then added to liquid soft agar (0.4% *w*/*v*) and poured onto a solid agar plate. After overnight incubation at 37 °C, clear plaques indicating the presence of lytic phages were harvested using a pipette tip and resuspended in an SM-buffer (100 mM NaCl, 50 mM Tris-HCl, 8 mM MgSO_4_ and 0.01% gelatin). The suspension was mixed with the log-phase growing *E. coli* propagating strain in liquid soft agar and poured onto a solid agar plate. Clear plaques were suspended in an SM buffer after overnight incubation followed by streaking on a double agar plate. At least 4 consecutive single plaque isolates were processed for a pure phage preparation, which was used for further propagation. In general, logarithmic growing cultures were infected with phages and incubated at 37 °C 180 rpm until visible lysis or overnight. After centrifugation and filtration (ULTRAFREE-CL GV 0.22 µm Centrifugal Devices (Merck Millipore, Germany)), the titer of the lysate was determined using the double agar method and the lysate was stored at 4 °C.

### 4.3. Transmission Electron Microscopy (TEM)

For TEM analysis, phages were treated as previously described [57]. Briefly, phages were allowed to adsorb onto thin carbon support films. Afterwards, they were negatively stained with 2% (*w*/*v*) aqueous uranyl acetate, pH 5.0. Samples were examined in a Zeiss EM-910 or Zeiss Libra120Plus transmission electron microscope (Carl Zeiss, Oberkochen, Germany) at an acceleration voltage of 80/120 kV at calibrated magnifications using a crossed line grating replica. Size determination of heads and tails was performed using the ITEM Software (Olympus Soft Imaging Solutions, Münster, Germany) from at least 3–10 different phage particles and further classification was conducted according to Ackermann [58].

### 4.4. Plaquing Host Range (PHR) and Efficiency of Plaquing (EOP)

Logarithmic growing bacterial cultures were diluted in liquid soft agar (0.4%) to a concentration of ~1 × 10^7^ CFU/mL. The soft agar was poured onto rectangular solid agar plates. After solidification, 3 µL of SM buffer serial-diluted phage lysates were dropped onto the bacteria-containing layer. Plates were kept at 37 °C and plaques were counted after several hours or overnight incubation. EOP was calculated as the ratio of the PFU measured on the strain under investigation and the host strain was used for propagation. The PHR was defined as the percentage of strains showing clear plaque formation. Strains with no clear plaque formation but impairment of growth at the site of the phage spot (“opaque lysis”) were not counted as plaque forming. PHR and EOP determinations were performed twice.

### 4.5. Kinetic Host Range (KHR)

Bacterial overnight cultures and phage lysates were diluted in fresh artificial urine to approximately 2 × 10^7^ CFU/mL and 2 × 10^8^ PFU/mL, respectively, and mixed 1:1 (= MOI 10) in a 96-well plate with a final reaction volume of 150 µL. Bacteria without added phages were used as untreated control. Plates were incubated overnight at 37 °C in a humidified atmosphere and OD_600_ was measured after 6 and 24 h incubation using the Multiskan^™^ FC microplate photometer (Thermo Scientific, Waltham, MA, USA). The KHR was defined as the percentage of strains showing no growth after 24 h. Therefore, OD-ratios between untreated control and phage-treated wells were calculated and an OD ratio of < 0.1 was defined as no growth. KHR was determined twice.

### 4.6. Phage DNA Isolation

Phage DNA isolation was carried out with the Phage DNA Isolation Kit (Norgen Biotek, Thorold, ON, Canada) or with DNeasy Blood & Tissue Kit (Qiagen, Hilden, Germany) based on the instructions provided by the manufacturer, with slight modifications. Phage lysates were concentrated, if necessary, prior to DNA extraction by precipitation with 10% PEG 8000 and 1M NaCl as described [59] to ensure concentrations of ≥ 10^8^ PFU/mL. Prior to DNA isolation, phages lysates of at least 10^8^ PFU/mL were DNase and RNase treated to avoid contaminants from the bacterial host. The DNase was then inactivated by adding EDTA and heating up to 65 °C. For the Phage DNA Isolation Kit, Lysis Buffer B was added, and the sample was vortexed vigorously for 10 s, followed by 15 min incubation at 65 °C, mixing the tube by inversion every 5 min. After adding isopropanol, the sample was flown through the provided column and allowed to bind. The column was washed by adding Wash Solution A and centrifuging at 6000× *g* for 1 min. To elute the DNA, the column was incubated for 1 min with Elution Buffer B and centrifuged for 1 min at 6000× *g*. For the DNeasy Blood & Tissue Kit, samples were treated with Proteinase K for 30 min at 55 °C before adding buffer AL, mixing thoroughly, and incubating for 10 min at 70 °C. After adding ethanol, the sample was flown through the provided column and allowed to bind. The column was washed by adding buffer AW1 and subsequent AW2 and centrifuging at 6000× *g* for 1 min. To elute the DNA, the column was incubated for 1 min with nuclease-free water and centrifuged for 1 min at 6000× *g*. Samples were stored at 4 °C. The concentration was determined using the Qubit^®^dsDNA HS Assay Kit (Thermo Fisher Scientific) following the manufacturer’s instructions.

### 4.7. Library Preparation, Whole Genome Sequencing and Genomic Analysis

An amount of 0.5 ng DNA was enzymatically fragmented using the Nextera DNA Library Preparation Kit (Illumina, FC-121-1030) for 10 min at 55 °C. After the addition of 11 μL KAPA-master mix (Library Amplification Kit, Kapa Biosystems) and 4.4 μL of each Nextera index primer, the reaction mixture was incubated according to the following program: 3 min at 72 °C, 5 min at 98 °C, 13 × (10 s at 98 °C, 30 s at 62 °C, 30 s at 72 °C), and 5 min at 72 °C. For PCR clean-up and size selection, the PCR product was incubated with magnetic beads (Ampure XP, Beckman Coulter). The pool of DNA libraries was adjusted to 4 nM and sequenced using MiSeq technology (Reagent Kit v3, 600-cycle, MS-102-3003, Illumina).

Before assembly, the sequencing reads were trimmed using prinseq-lite 0.20.4 [60]. Quality checks of the reads were performed with FastQC 0.11.9 [61]. SPAdes genome assembler version 3.13.0 and 3.14.1 [62] was used for the assembly of Illumina reads. Read mapping was conducted with the map to reference feature of Geneious (Geneious Prime 2020.01.02.) with medium sensitivity. Phage genome annotation was conducted with Prokka version 1.14.5 [63].

Analysis of genome mosaicism was conducted by comparing each CDS sequence of the bred phages with the ancestor sequence, determining the best matching ancestor. This was conducted manually using the progressive Mauve algorithm in build into Geneious.

Taxonomic classification was inferred from the closest sequenced relatives identified by BLAST analysis of phage sequences to NCBI’s RefSeq database.

The lytic lifecycle of the ε^2^-phages was confirmed computationally using PHACTs [64]. Antibiotic-resistance genes were analyzed by uploading the translated sequences to Res-Finder 4.1 [65], using parameters for *E. coli* species, targeting chromosomal point mutations, and acquired antimicrobial resistance genes (60% minimum length, 90% threshold ID, database version 2021-04-13). The absence of virulence factors was analyzed by BLASTp, against the Comprehensive Antibiotic Resistance Database (CARD, v3.1.0 of April 2021) [66]. The absence of virulence genes was further confirmed with Virulence Finder 2.0 CGE, [67,68] against *E. coli* (60% minimum length, 90% threshold ID, database version 2020-05-09).

### 4.8. Breeding of Phages

Phage breeding was conducted based on the principles of the Appleman’s Protocol [23]. Briefly, 3 to 4 ancestor phages were pooled to create the input phage mixture which was used (diluted and undiluted) to infect each of the subpanels of 24 *E. coli* strains in a 96-well microtiter plate. Two separate phage combinations were used, a mixture of the Myoviruses CHD16UKE1, CHD94UKE2, 101117BS1, and G3G7, and a mixture of members of the *Autographiviridae*, 22664UKE3-2, 101118B1, and 101120B2. For the first round of breeding, single phages with similar concentrations were mixed and then serial 10-fold diluted in artificial urine. Starting concentrations were 2 × 10^7^ PFU/mL for myoviruses and 2 × 10^6^ PFU/mL for *Autographiviridae*. Bacterial overnight cultures were diluted in fresh artificial urine to ~1 × 10^8^ CFU/mL. Bacteria and phage dilution volumes were mixed 1:1. After overnight incubation at 37 °C, clear lysates and the first dilution of turbid lysates were pooled, sterile filtrated, and used to infect the next round of breeding. This was repeated for up to twenty rounds. Single individual phages were isolated from the bred mixture lysate by re-streaking on the host strain at least 4 times. 

### 4.9. Determination of Phage Efficiency

Phage virulence was examined by determination of the bacterial growth after incubation of a constant phage concentration with different numbers of bacteria. In detail, 22 *E. coli* strains with a starting inoculum of 1 × 10^5^, 1 × 10^6^, 1 × 10^7^ and 1 × 10^8^ CFU/mL were mixed with 1 × 10^7^ PFU/mL (MOI 100, 10, 1 & 0.1) of each ancestor and selected ε^2^-phages in artificial urine in a 96-well plate. Plates were incubated overnight at 37 °C and OD_600_ was measured after 24 h incubation. OD-ratios between the untreated control and phage-treated wells were calculated and an OD ratio of <0.1 was defined as no growth. Determinations were performed in triplicate.

### 4.10. Statistics

One-and two-way Anova analysis were performed with GraphPad Prism 9.

## Figures and Tables

**Figure 1 antibiotics-10-01337-f001:**
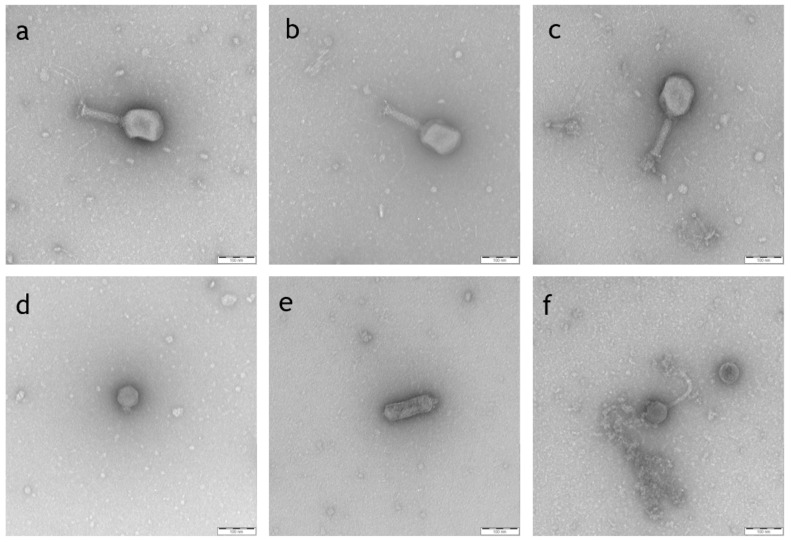
Electron micrographs of isolated phages belonging to different families. Negative staining (2% (*w*/*v*) uranyl acetate, pH 5.0) of *Myoviridae* phage CHD94UKE2 (**a**), 101117BS1 (**b**) and CHD16UKE1 (**c**); *Autographiviridae* phage 10112B02 (**d**), *Podoviridae* phage 101114BS3 (**e**) and *Siphoviridae* phage 22664BS2 (**f**). Bars represent 100 nm.

**Figure 2 antibiotics-10-01337-f002:**
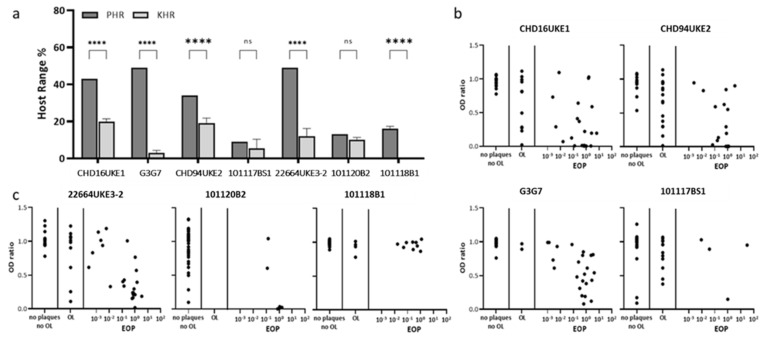
Host range of selected phages on UPEC panel. (**a**) Comparison of plaquing (PHR) and kinetic (KHR) host range over the panel of 47 *E. coli* strains (Table 2). Shown are mean values + SD. (**b**,**c**) Correlation of OD_600_ ratio after 24 h and EOP for tested *Myoviridiae* (**b**) and *Autographiviridae* (**c**). ns—not significant, **** *p* < 0.0001.

**Figure 3 antibiotics-10-01337-f003:**
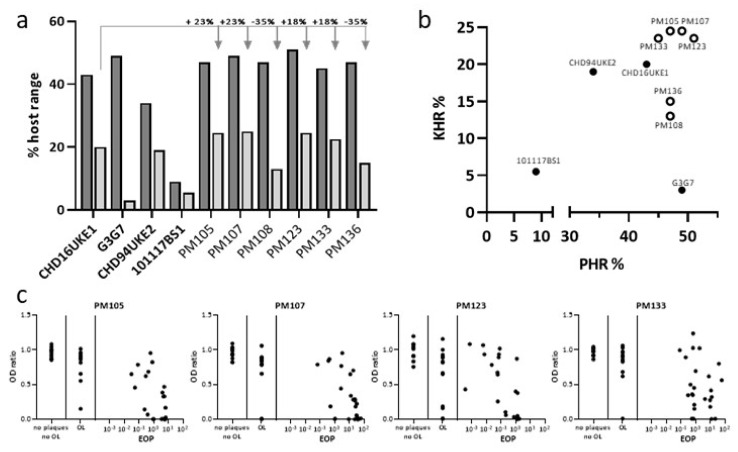
Host range of ε^2^-phages. (**a**) Comparison of plaquing (PHR) and kinetic (KHR) host range over the panel of 47 UPEC strains for ancestor (bold) and ε^2^-phages. (**b**) Correlation of PHR and KHR of ancestor (closed circle) and ε^2^-phages (open circle). (**c**) Correlation of OD_600_ ratio after 24 h and EOP for ε^2^-phages.

**Figure 4 antibiotics-10-01337-f004:**
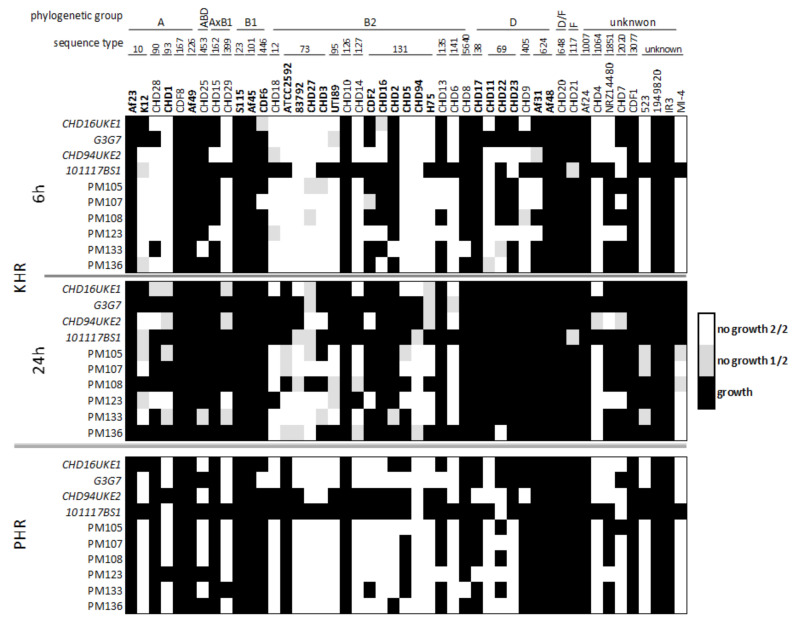
Heat map of host range measurement. *E. coli* strains (horizontal axis) are grouped according to their phylogenetic group and sequence types. The 24 strains used for the breeding are marked bold. Phages (vertical axis) are grouped by type of host range measurement (KHR–kinetic host range, PHR–plaquing host range), ancestor phages (italic) above ε^2^-phages. Black squares indicate failure to inhibit growth in suspension (KHR) or absence of plaques (PHR) in 2/2 replicates, grey squares indicate no growth in one of two repetitions and white squares indicate no growth or presence of plaques in both repetitions.

**Figure 5 antibiotics-10-01337-f005:**
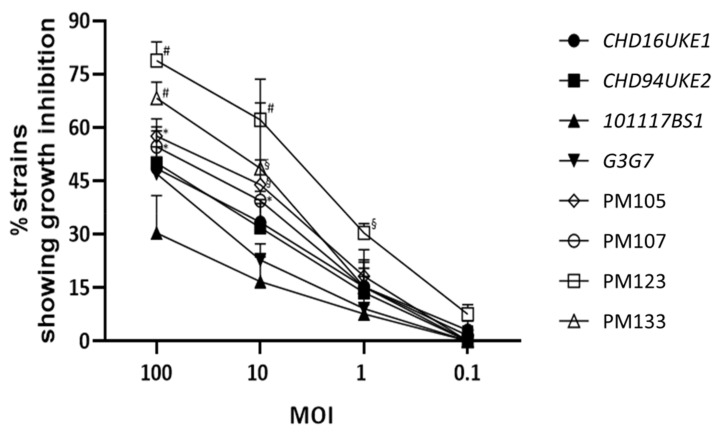
*Myoviridae* ε^2^-phages were more virulent than their ancestors. Ancestor (italic and dark-fill symbols) as well as selected ε^2^-phages (no-fill symbols) with an inoculum of 1 × 10^7^ PFU/mL were used to infect different bacterial inocula (1 × 10^5^ − 1 × 10^8^ CFU/mL = MOI 100–0.1) of 22 UPEC strains. Bacterial growth was measured after 24 h and growth inhibition was defined as an OD_600_ ratio over the uninfected control <0.1. *—*p* < 0.05 over 101117BS1, ^§^—*p* < 0.05 over 101117BS1 & G3G7, ^#—^*p* < 0.05 over all 4 ancestors.

**Figure 6 antibiotics-10-01337-f006:**
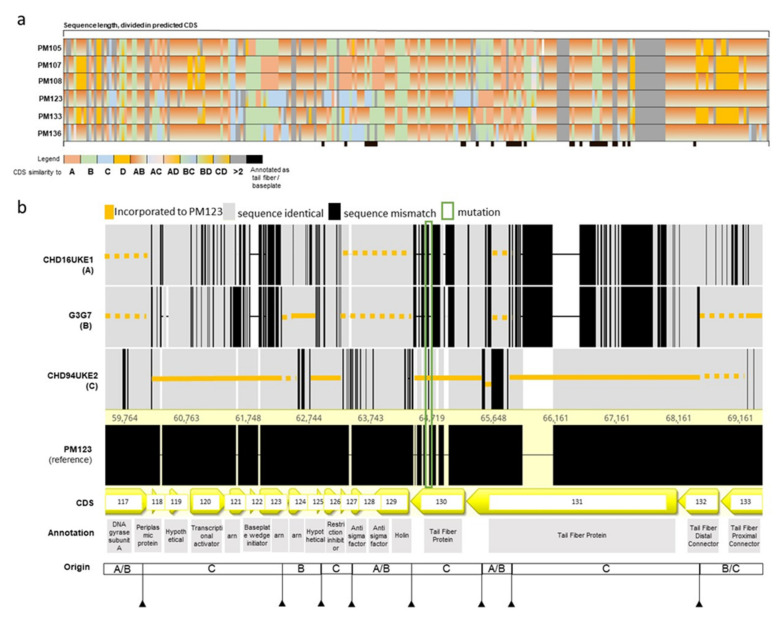
Sequence analysis of ε^2^-phages. (**a**) Comparison of the mosaicism of ε^2^-phages belonging to the *Myoviridae* family. Phages are aligned against PM123 and genome sequence were divided in the predicted CDS (1 to 279), and each CDS is given a color according to sequence similarity with the ancestors. As indicated in the legend, the phage CHD16UKE1 (A) is represented in pink, G3G7 (B) in green, CHD94UKE2 (C) in blue and 101117BS1 (D) in orange. Mixed colors indicate two possible ancestors. The grey CDS represent a sequence similarity to more than two ancestors, while white CDS represents the lack of relation to any of the ancestors. CDS with a predicted role in tail or baseplate formation are marked in black in the last column. (**b**) ~10 kbp genome region of PM123, which is a mosaic of ancestor phages CHD16UKE1 (A), G3G7 (B) and CHD94UKE2 (C). Dashed lines indicate similarity to both ancestors. Arrows indicate recombination sites. Annotation of predicted CDS is indicated, referring to its number. Arn—putative anti-restriction nuclease.

**Table 1 antibiotics-10-01337-t001:** Phages isolated from wastewater samples.

Order	Family	Subfamily	Genus	Phage Name	Genome Length bp	Host Strain: *E. coli* ^#^	Accession Number	HR * [%]
Caudovirales	*Autographiviridae*	*Molineuxvirinae*	*Vectrevirus*	101101UKE1	44450	DSM101101	MZ234012	19
				UTI89UKE2	44293	UTI89	MZ234049	3
				UTI89UKE3	44294	UTI89	MZ234050	4
			unspecified	101117UKE2	44526	DSM101117	MZ234019	8
				101118B1	44526	DSM101118	MZ234020	51
		*Studiervirinae*	*Kayfunavirus*	101118UKE1	40233	DSM101118	MZ234021	41
				101120B1-2	39845	DSM101120	MZ234022	19
				101120B2	39899	DSM101120	MZ234023	34
				101136BS1	39375	DSM101136	MZ234024	44
				22664BS1	39133	DSM22664	MZ234009	42
				22664UKE3-2	40482	DSM22664	MZ234011	61
	*Myoviridae*	*Tevenvirinae*	*Mosigvirus*	172859UKE1	168667	172859	MZ234025	4
			*Tequatrovirus*	101112UKE3-1	169555	DSM101112	MZ234013	37
				101117BS1	167080	DSM101117	MZ234018	23
				CHD16UKE1	168543	CHD16	MZ234030	42
				CHD2BS1	168577	CHD2	MZ234027	13
				CHD94UKE2	167922	CHD94	MZ234031	29
				G3G7	168649	CHD16	MZ234040	n.d.
	*Podo-viridae*	[x]	*Kuravirus*	101114BS3	75747	DSM101114	MZ234015	53
				101114UKE3	75747	DSM101114	MZ234017	32
				CHD5UKE1	77359	CHD5	MZ234028	1
	*Siphoviridae*	[x]	*Dhillonvirus*	101114B2	44971	DSM101114	MZ234014	38
				101114BS4	45251	DSM101114	MZ234016	27
				22664B1	45019	DSM22664	MZ234008	41
				CHD2B1	45144	CHD2	MZ234026	15
				CHD5UKE2	45243	CHD5	MZ234029	24
				UTI89UKE1	41265	UTI89	MZ234048	1
		*Guernseyvirinae*	*Kagunavirus*	22664BS2	45176	DSM22664	MZ234010	43

[x]—no subfamily assigned. ^#^—all strains, except for UTI89 (ST95, O18) and 172859 (ST unknown, O25b), are O25b-antigen positive (identified by PCR as described by [42]), belonging to the 131 sequence type (DSM-strains—ST131 affiliation is assumed on the basis of the O25b antigen, but has not been proven). *—plaquing host range percentage measured against a panel of 79 *E. coli* strains originating from UTI and representing different serotypes (see Appendix A).

**Table 2 antibiotics-10-01337-t002:** Overview of sequence types of *E. coli* strains used for further characterization. A comparison between the % of strains in this study and the literature range is shown.

Sequence Type	Phylogenetic Group	Number of Isolates Used in This Study	% of Total Strains in This Study	Literature ^a^ Range % of Strains
ST131	B2	6	13%	7.7–29
ST95	B2	1	2%	3–28.9
ST73	B2	3	6%	8.8–11
ST69	D	3	6%	3–9
ST38	D	1	2%	1.2–7
ST10	A	2	4%	1.8–6.4
ST127	B2	1	2%	0.6–5.9
ST141	B2	1	2%	4.4
ST12	B2	1	2%	1.2–3.8
ST405	D	1	2%	0.6–3
ST624	D	2	4%	1.5
ST23	A	1	2%	1.5
ST93	A	1	2%	1,2
ST101	B1	1	2%	1
ST167	A	1	2%	1
ST162	AxB1	1	2%	0.6–1
ST2020	n.d.	1	2%	0.5
ST90	A	1	2%	n.d.
ST399	A (AxB1)	1	2%	n.d.
ST226	A0	1	2%	n.d.
ST453	ABD	1	2%	n.d.
ST446	B1	1	2%	n.d.
ST126	B2	1	2%	n.d.
ST135	B2	1	2%	n.d.
ST5640	B2	1	2%	n.d.
ST648	D/F	1	2%	n.d.
ST117	F (B1/D)	1	2%	n.d.
ST1007	n.d.	1	2%	n.d.
ST1064	n.d.	1	2%	n.d.
ST1851	n.d.	1	2%	n.d.
ST3077	n.d.	1	2%	n.d.
Not determined	n.d.	5	11%	n.d.

^a^ [2,3,4,5,6,7], n.d.—not determined.

## Data Availability

The data presented in this study are available within the article and supplementary material.

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
