# Peer review of "Natural Bred ε2-Phages Have an Improved Host Range and Virulence against Uropathogenic Escherichia coli over Their Ancestor Phages"

_antibiotics, 2021, doi:10.3390/antibiotics10111337_

Round 1
Reviewer 1 Report
The manuscript ‘Natural bred ε2-phages have an improved host range and virulence against uropathogenic Escherichia coli over their ancestor phages’ by Loose, et al., demonstrates that wild-type bacteriophages could be bred by sequential cultivation in artificial urine into evolution-squared-phages to gain a wider host range and stronger virulence against uropathogenic E. coli compared to their ancestor phages. The study is well constructed, and the data presented assures a promising future for bacteriophages in antibacterial drug discovery.
Minor comments:
- Results
2.2 Page 6:
Do the two sentences below describe Figure 2a? The only Myoviridae in Figure 2a are CHD16UKE1, CHD94UKE2, and 101117BS1, therefore there is no data presented in Figure 2a for G3G7, as described in the text. Additionally, data for Autographiviridae 22664BS1 is presented in Figure 2a, but there is no description for this phage in the text.
Lines 21-24 ‘The myovirus G3G7 as well as the Autographiviridae 22664UKE3-2 showed the highest PHR with each 49% followed by ------ (Figure 2)’;
Lines 32-35 ‘Phages G3G7 and 22664UKE3-2 showed a particularly large difference between PHR and KHR, ----- (Figure 2)’.
Author Response
Thank you very much for pointing out at this error. There was a mistake in the labelling of the graph, which has now been corrected.
Reviewer 2 Report
Materials and methods:
In sections 4.2. and 4.5, the MOI used for the assays should be explicit.
In section 4.5., it should be mentioned where the OD600 nm measurements were performed.
In section 4.6., please clarify or rephrase the sentence “In some instances, phages were concentrated by precipitation…”
Discussion:
In this manuscript, the authors show that ε2-phages lysing E. coli, bred from newly isolated wild type phages, demonstrate higher KHR and virulence against E. coli strains, comparing with the wild type phages. However, comparing with a recent paper cited by the authors (Moreno et al., 2021), where the same strategy is used for S. aureus phages, results are not as striking. It would be useful to better discuss how host range and virulence could be improved in ε2-phages of such diverse E. coli strains, so that ε2-phages could be more realistically considered for phage therapy against UPEC.
Author Response
Thank you for the constructive comments, we have updated the sections accordingly.
In section 4.2 Phage isolation and propagation, the MOI is not stated since the viral particles present in wastewater is unknown.
Reviewer 3 Report
I have no objection to the Manuscript. All elements were carried out and described correctly. Minor editorial corrections required. In Table 2, Supplement 2, the name of the antibiotic - nalalidixic acid should be changed to English in the caption (it is in German). Figure 2 is too small in the text , illegible, it is worth enlarging the charts. On the other hand, the Table 2 can have a smaller font.
Author Response
Thank you for your constructive comment. We have changed supplement table 2 accordingly.